# In Situ Corn Fiber Conversion for Ethanol Improvement by the Addition of a Novel Lignocellulolytic Enzyme Cocktail

**DOI:** 10.3390/jof8030221

**Published:** 2022-02-24

**Authors:** Le Gao, Dongyuan Zhang, Xin Wu

**Affiliations:** Tianjin Key Laboratory for Industrial BioSystems and Bioprocessing Engineering, National Technology Innovation Center of Synthetic Biology, Tianjin Institute of Industrial Biotechnology, Chinese Academy of Sciences, No. 32, Xiqi Road, Tianjin Airport Economic Park, Tianjin 300308, China; zhang_dy@tib.cas.cn

**Keywords:** lignocellulolytic enzymes, corn mash, viscosity decrease, ethanol yield, DDGS

## Abstract

Corn mashes have high-viscosity and high-sugar characteristics, which hinders yeast-fermentation efficiency and the ethanol yield increase. The excessive viscosity of corn mash is caused by the unutilized cellulose in corn kernel fiber. A novel lignocellulolytic enzymes cocktail with strong substrate specificity was prepared for high-viscosity, high-sugar corn mash. The in situ conversion of corn mashes with novel lignocellulolytic enzymes at the optimum cellulase dosage of 50 FPU/L resulted in about 12% increased ethanol concentration compared with the reference mash at different batch-fermentation scales. Adding the lignocellulolytic enzymes caused the greatest decrease in viscosity of corn mash and residual sugars by 40.9% and 56.3%, respectively. Simultaneously, the application of lignocellulolytic enzymes increased the value of the dried distiller’s grain with solubles (DDGS) by increasing the protein content by 5.51%. The in situ conversion of cellulose can decrease the fermentation broth viscosity and improve the rheological property, thereby improving the ethanol yield. With the same amount of material, the application of the novel enzymes cocktail can enhance the ethanol yield by more than 12%. A quarter of the ethanol yield increase was due to the further hydrolysis of starch, while three quarters to cellulose. Thus, this technology will increase the net revenue of bioethanol industrialization.

## 1. Introduction

Ethanol production through biotechnology is an attractive option for sustainable fuel production. Starch is the second most important and abundant source of carbon and energy in plants. Starch is an important feedstock in the fermentation industry and is widely fermented to produce ethanol, which can be used as a basis for beverages or as an alternative biofuel [1,2]. In China, the major feedstock for ethanol fuel production is corn, and new production plants tend to apply the dry milling technology more than wet milling because of lower capital costs [3]. In the dry milling process, whole corn kernels are milled and mixed with water, yielding a viscous slurry, which is liquefied by heat treatment and α-amylase. The economics of this process greatly depends on the revenue obtained by selling the main co-product known as dried distiller’s grain with solubles (DDGS). It is rich in protein, fiber, and vitamins as an animal feed [4]. Attempts have been made to develop additional co-products, such as corn oil [5]. 

Industrial corn mash has high viscosity and contains coarse particles. A problem during the processing of corn starch is the excessive viscosity of mash, which can increase the complexity of the process of starch hydrolysis and mash fermentation [6]. The excessive viscosity of mash is caused by the presence of non-starch polysaccharides, such as cellulose and hemicellulose in cereal grains [6]. The corn kernel fiber (mostly cellulose) is unutilized in the current process. Corn cellulose conversion is of great interest in the field [7]. However, most studies focused on the conversion of isolated corn fiber [8,9]. The in situ conversion of cellulose during the dry-milling process has not been studied comprehensively. Corn consists of an outer seed cover or pericarp, which encloses the embryo and the starch-rich endosperm [5]. This structure renders them non-hydrolyzable by amylolytic enzymes to fermentable sugars. The lignocellulolytic material can serve as feedstock for additional ethanol production. The recalcitrance of corn starch requires lignocellulolytic enzymes preparation, resulting in an unacceptable and costly process.

The reported work showed lignocellulolytic enzymes could improve ethanol yield from starch due to excessive enzyme dosage. The costly process brings out an inappropriate ethanol yield increase, which is unacceptable and hinders cellulase application in the field of bioethanol. 

In this work, we develop a novel lignocellulolytic enzymes preparation that has strong substrate specificity for high-sugar and high-viscosity corn mash. The main objectives are to convert cellulose in situ during simultaneous saccharification and fermentation (SSF) and to investigate the effect of the novel lignocellulolytic enzymes addition on SSF performance, including ethanol yield, corn mash viscosity, and residual sugars. The application of the novel enzymes cocktail can enhance ethanol yield by more than 12%, which is beneficial to bioethanol.

## 2. Materials and Methods

### 2.1. Mash Preparation 

Corn mash preparation was provided by Jilin Fuel Alcohol Company Limited (Jilin, China). Equal amounts of tap water and back-set stillage were mixed with ground corn to a final dry matter content of 27% (*w*/*v*). 

### 2.2. Culture Condition

The method of lignocellulolytic enzyme fermentation was referring to the reported [10,11]. For strain recovery, mycelia agar disks were inoculated on fresh PDA and cultivated at 30 °C for 7 days until conidia formed. Exactly 1 mL of spore suspension was incubated in 30 mL of preculture (glucose 10 g/L, corn steep powder 10 g/L, pH 5.0) in a 250 mL flask and cultured at 28 °C and 180 rpm for 24 h. Then, 5% inoculum size of preculture was transferred to 30 mL of production medium in a 250 mL flask at 26 °C and 180 rpm for 5 days. The production medium comprised 17 g/L corn steep powder, 5 g/L (NH_4_)_2_SO_4_, 6 g/L KH_2_PO_4_, 1 g/L MgSO_4_∙7H_2_O, 2.5 g/L CaCO_3_, 2 mL/L tween-80, and 30 g/L inducers. It had an initial pH of 5.0. 

### 2.3. Enzyme Preparations

#### 2.3.1. Cellulase from *T. reesei*


*T. reesei* A2H (China General Microbiological Culture Collection Center, Beijing, China; CGMCC 21470) was a lignocellulolytic enzyme-hyperproducing mutant strain obtained by chemical mutagenesis in our laboratory [12]. The medium used for the lignocellulolytic enzyme production was prepared in accordance with a previous study [12].

#### 2.3.2. Auxiliary Enzymes Expressed in *A. niger*


*A. niger* 60B-3DW (China General Microbiological Culture Collection Center; CGMCC 22465) was a β-glucosidase hyper-producing mutant strain obtained by chemical mutagenesis in our laboratory. To clone the whole fragment of the FAE (GenBank accession number: KY449280) and GH61 (GenBank accession number: ATQ35955.1) encoding gene from the chromosomal DNA of *Penicillium piceum* 9-3, a pair of oligonucleotide primers was designed based on the sequenced genomic data of *P. piceum* 9-3. The FAE and GH61 gene were digested with PstI and SalI and then separately inserted into digested p19-Pptef-2Apeptide-hph-Tcbh1 to generate enzyme-expression plasmids. Transformation of *A. niger* protoplasts was performed according to the previously described procedure [13]. The transformants were selected on plates with Hygromycin B via single spore culture for further analysis. Positive clones were verified through DNA sequencing. 

#### 2.3.3. Lignocellulolytic Enzymatic Preparation Enzymes

The novel lignocellulolytic enzymatic preparation was a cocktail with cellulase from *T. reesei* and auxiliary enzyme from *A. niger* at the ratio of 8.5:1 (*v*/*v*). 

### 2.4. Yeast Preparation 

Approximately 1.0% of alcohol instant active dry yeast (*S. cerevisiae*; Angel, Yichang, China) was suspended in distilled water at 32 °C for 1 h. Then, 1.0 mL of yeast was added into 100 mL of corn mash with a certain quantity of lignocellulolytic enzymes.

### 2.5. Simultaneous Saccharification and Fermentation (SSF) of Corn Mashes

Following yeast and enzyme additions, the corn mash was stirred thoroughly to ensure uniform distribution of the enzyme and yeast. The SSF of corn mashes was conducted for 36 h at 32 °C. The fermentation tanks were capped with rubber stoppers that were pierced with needles to allow for the release of CO_2_. The fermentation tanks were sparged with oxygen at 0.6 L/min. During this time, the flasks/tanks were periodically weighed to determine the weight loss due to CO_2_ production. The weight loss data were used to confirm that all fermentations were complete at 36 h. The samples of corn mashes were taken and centrifuged on a microcentrifuge. Final samples were filtered through a 0.22 µm micron filter to detect the fermentation parameters, such as ethanol content and reducing sugars. 

### 2.6. Degradability of Residual Starch and Residual Cellulose

After SSF for 36 h, the broth without any treatments was used as the control group to study the degradability of residual starch and cellulose. To remove the ethanol effect on the cellulase activity, ethanol was evaporated from the SSF broth with a rotary evaporator at 65 °C, for 15 min, and at 0.07 MPa. The residual starch and cellulose were degraded in a 100 mL Erlenmeyer flask with the same substrate concentration of 27%. The lignocellulolytic enzymes were then added to hydrolyze at 50 °C for 40 h. Then, the yeast was added to the broth for ethanol fermentation. The residual starch and cellulose contents were then analyzed again.

### 2.7. Analysis of Fermentation Samples

The corn mash samples were analyzed. The reducing sugars and total sugar concentrations after acid hydrolysis were determined (both expressed in grams glucose per 100 mL mash) according to the Schoorl and Regenbogen method [6]. The dextrins were calculated as the difference between the total and reducing sugars, considering the conversion coefficient into dextrins (0.9) and expressed in 100 mL mash. 

The starch contents of the maize mash were determined using the modified Megazyme assay. The method was based on the hydrolysis of starch with α-amylases and glucoamylase to produce glucose, which was subsequently calculated to the starch content of the sample. 

The carbohydrate in biomass was quantitatively analyzed according to the NREL Laboratory Analytical Procedures (NREL, 2006) for biomass using a two-step acid method [14]. Approximately 1 g (dry basis) of the samples was dispensed into 200 mL Erlenmeyer flasks. The samples were treated with 5 mL of 72% (*w*/*w*) H_2_SO_4_ at 30 °C for 2.5 h and then stirred every 15 min with a glass stirring rod. The solutions were diluted with 181.7 mL of water and then autoclaved at 121 °C for 1 h. Glucose and xylose concentrations were determined through high-performance liquid chromatography (HPLC) (Shimadzu, Kyoto, Japan) with a refractive index detector (Shimadzu) on an Aminex HPX-87H column (Bio-Rad, Hercules, CA, USA) running at a flow rate of 0.6 mL/min at 60 °C, with 5 mM H_2_SO_4_ as the mobile phase. Glucan and xylan concentrations were calculated according to Equations (1) and (2), where the factors of 0.9 and 0.88 reflect the weight loss as a result of glucose-to-glucan and xylose-to-xylan conversion, respectively [15].
(1)$$\mathrm{Glucan}\;\mathrm{content}\;(\%)=\frac{\mathrm{Glucose}\;\mathrm{released}\;\mathrm{from}\;\mathrm{acid}\;\mathrm{hydrolysis}\;(\mathrm{mg})\times 0.9}{\mathrm{Samples}\;\mathrm{weight}\;(\mathrm{mg})}\times 100\%$$
(2)$$\mathrm{Xylan}\;\mathrm{content}\;(\%)=\frac{\mathrm{Xylose}\;\mathrm{released}\;\mathrm{from}\;\mathrm{acid}\;\mathrm{hydrolysis}\;(\mathrm{mg})\times 0.88}{\mathrm{Samples}\;\mathrm{weight}\;(\mathrm{mg})}\times 100\%$$

All samples were diluted and passed through a 0.45 µm filter before HPLC analysis. Residual glucose, ethanol, and other fermentation minor products were determined using high-performance liquid chromatography (HPLC, Shimadzu, Kyoto, Japan) with a refractive index detector (Shimadzu) on an Aminex HPX-87H column (Bio-Rad, Hercules, CA, USA). The flow rate was 0.6 mL/min at 55 °C, and the mobile phase was 5 mM H_2_SO_4_. Each sample was injected three times, and the average results were calculated [16].

The viscosity was determined using a Hoppler viscometer type Visco Ball, expressed in mPa s at 20 °C. The determination principle for the viscosity of liquids using the Hoppler viscosimeter involved the measurement of the ball descent time through a constant distance in the studied liquid contained in a glass tube.

According to the Kjeldahl method, 1 g samples are placed in the digestion tube. Twenty milliliters of sulfuric acid and a tablet containing 0.48 g of mercury oxide as a catalyst and 4.52 g of potassium sulfate are added. Blanks containing all these reagents are simultaneously processed. The tubes are placed in the preheated digestion block at 420 °C for 2 h 30 min. The resulting solutions are cooled at room temperature and diluted by adding 30 mL of water. The tubes are placed in the distillation–titration unit. Then, 20 mL of sodium hydroxide solution are automatically added and the solutions are distilled (6 min). The ammonia collected in the receiving solution (30 mL) is automatically titrated against the standard 0.25 M hydrochloric acid with colorimetric endpoint detection. Ammonium dihydrogen phosphate standard (NIST SRM 194) is used to check the concentration of the titrating solution. The total N content was calculated according to ammonium dihydrogen phosphate content [17]. Crude protein content was calculated using total *n* content data (N × 6.25) [6]. The reducing sugars were determined by the DNS method according to the procedures of Song et al. [18]. 

### 2.8. Statistical Analysis

For comparison of DDGS composition with/without lignocellulolytic enzymes addition, ANOVA was used, followed by Tukey’s multiple-comparison posttest with a 95% confidence interval. Statistics were performed by using SigmaPlot (version 11) [19]. The differences were considered significant at *p*  <  0.01 [20]. 

## 3. Result and Discussion

### 3.1. Characteristics of Lignocellulolytic Enzymes

The composition of the lignocellulolytic enzyme cocktail for high-viscosity corn mash would be screened in the lignocellulolytic enzyme library. The lignocellulolytic enzyme library contained the main enzyme system from *T. reesei* and the coenzyme system from *A. niger*. Although the commercial cellulase was produced from *T. reesei*, the enzyme system from *T. reesei* has some drawbacks [21]. The main enzyme system from *T. reesei* was incomplete and unbalanced. That is, it had no genes encoding cellobiose dehydrogenases and feruloyl esterase and few monooxygenases and β-glucosidase expressed in the *T. reesei* extracellular protein [22,23]. These key coenzymes may need to be introduced from other species to improve the enzymatic efficiency [21]. For high-viscosity, high-sugar corn mash, the coenzymes such as FAE, GH61, and β-glucosidase were overexpressed up to 15% of total extracellular enzymes for optimization of enzyme system composition. It is reported that this FAE, GH61, and β-glucosidase from *Penicillium piceum* had unique enzymology characteristics in our previous work [24,25]. The FAE and GH61 reduced biomass recalcitrance by increasing the xylose/arabinose ratio and decreasing the HBI of crude biomass. This could reduce lignocellulose degradation recalcitrance, providing favorable conditions for enzymatic hydrolysis [24,25]. The novel lignocellulolytic cocktail had a balanced enzymatic ratio with filter paper activity of 40.8 FPU/mL, xylanase of 1320.6 IU/mL, feruloyl esterase of 22.5 IU/mL, and additional side activities, including amylase of 4.5 IU/mL.

### 3.2. Effect of Lignocellulolytic Enzyme Loadings on Corn Mash Fermentation

Considering the cost of using enzymes, the lignocellulolytic enzyme was selected for further experiments to test at a lower enzyme dosage. Corn mash fermentation was performed at different enzyme loadings (2–200 FPU/L) to facilitate the screening of optimum enzyme dosage. The ethanol concentration, sugar reduction, and viscosity of corn mash corresponding to different enzyme dosages are presented in Table 1. The ethanol yield increased with increased cellulase dosage until 50FPU/L. Further increased enzyme dosage (60–200 FPU/L) did not increase the ethanol yield. The enzyme dosage ranges of 20–45 FPU/L did not result in maximal promotion in alcohol concentration and yield (Table 1). The ethanol yield reached an inflection on the enzyme dosage of 50 FPU/L (Appendix A). Therefore, 50 FPU/L was selected as the optimal cellulase dosage in the SSF. At the optimum enzyme dosage of 50 FPU/L mash, the ethanol yield increased by 13.3% compared with the reference mash.

The application of lignocellulolytic enzymes preparation decreased the mash viscosity and increased the yeast-fermentation efficiency. This result was consistent with a previously published report. The results of Czarnecki and Nowak indicate that there are beneficial effects of lignocellulolytic enzymes (e.g., xylanase, cellulase, and glucanase) on rye mashes, such as a decreased viscosity and an enhanced starch saccharification, and an increased productivity of ethanol [26,27]. 

### 3.3. Effect of Different Fermentation Tank Size on Ethanol Production

The novel lignocellulolytic enzymes were applied at different fermentation scales from 0.3–70 L. At the 0.3, 1, 5, and 70 L batch-fermentation scales. The reductions in corn mash viscosity were 46.3%, 31.6%, 35.5%, and 40.9% compared with the reference mash (33.5 ± 1.5 Pa·s), respectively. The treatment of corn mashes with the novel lignocellulolytic enzymes resulted in increasing concentrations of ethanol by 12.4%, 12.0%, 11.8%, and 12.9% compared with the reference mash. The highest yield of ethanol in the corn mash digested with the novel lignocellulolytic enzyme reached 117.0 ± 0.1 g/L at the 70 L batch fermentation, whereas that in the control reached only 103.6 ± 1.0 g/L. After the lignocellulolytic enzyme addition, the residual starch content decreased to 5.34 ± 0.26%, whereas that without lignocellulolytic enzyme was 6.58 ± 0.86% (Table 2). The residual cellulose content of corn mash decreased to 7.48 ± 0.10%, whereas that without lignocellulolytic enzyme was 12.64 ± 0.52%. Approximately 1.24% of starch and 5.16% of cellulose in the corn mash were further hydrolyzed. This result indicates that a part of the residual cellulose in corn mash was mostly degradable by the lignocellulolytic enzyme. The starch conversion with the cellulase cocktail improved because cellulase disrupted the cell wall structure of the grain and promoted starch release. Furthermore, the cellulase cocktail possibly contained amylases. 

### 3.4. Further Degradation of Residual SSF Broth after Ethanol Evaporated by Lignocellulolytic Enzymes Cocktail

To clearly elucidate the promotion effect of lignocellulolytic enzymes on alcoholic fermentation, the lignocellulolytic enzymes were added into the residual broth in SSF. The 36 h SSF broth (containing unconverted solids) with the ethanol evaporated was used for another round of separate hydrolysis and fermentation (SHF), which reduced the ethanol inhibition on the cellulase activity. As shown in Figure 1, approximately 7.0 g/L of glucose was released from the saccharification of the residual broth with the lignocellulolytic enzymes, whereas no glucose was released from that without the cellulase addition. Then, the activated yeast was added into the saccharification broth for another 36 h of ethanol fermentation. Further ethanol production can yield 6.44 g/L after lignocellulolytic enzymes addition. The residual starch contents with and without lignocellulolytic enzymes addition were 5.06 ± 0.34% and 6.02 ± 0.86%, respectively (Table 3). The residual cellulose contents of corn mash with and without lignocellulolytic enzymes additions were 10.07 ± 0.10% and 13.20 ± 0.52%, respectively. Approximately 0.96% of starch and 3.13% of cellulose in the corn mash were further hydrolyzed. Adding lignocellulolytic enzyme resulted in 13.58 g of sugar production, including the release of 1.43 g of soluble residual sugar and 12.15 g of glucose from starch and cellulose. These results indicate that the lignocellulolytic enzymes promoted the release of more glucose from residual starch and cellulose for ethanol increase but not yeast-fermentation efficiency. The hydrolysis efficiency of starch and cellulose in SHF was lower than that in SSF. 

### 3.5. Comparison of DDGS Composition

The main co-product of corn mash fermentation was DDGS, which can increase the economics of this process. A summary of DDGS composition is provided in Table 4. This result showed that lignocellulolytic enzymes addition exerted a significant effect (*p* < 0.01) on the composition of DDGS. The residual cellulose content of DDGS decreased (*p* < 0.01) from 33.12% to 21.20%, while the starch content of DDGS decreased (*p* < 0.01) from 14.07% to 11.26% (Table 4). These results showed that the lignocellulolytic enzymes addition promoted the further hydrolysis of residual starch and cellulose in corn mash. The crude protein in DDGS was determined to be 29.63% with the addition of the novel lignocellulolytic enzymes, and 24.12% without the novel lignocellulolytic enzymes. The value of DDGS, which is sold as animal feed, had a close relationship with crude protein content. Therefore, the application of lignocellulolytic enzymes increased the value of DDGS by increasing crude protein content by 5.51% (*p* < 0.01). No difference in DDGS color was observed (Appendix A).

### 3.6. Techno–Economic Analysis of the Strategy

The addition of novel lignocellulolytic enzymes for ethanol yield improvement does not require any additional equipment or control system. The only additional cost would be that of the lignocellulolytic enzymes. Ten tons of corn mash can produce one ton of ethanol. The ten tons of corn mash need added lignocellulolytic enzyme input of RMB 75 yuan (FPAase of lignocellulolytic enzyme calculated as 40 FPU/mL; RMB 6000 yuan/ton lignocellulolytic enzymes). The estimated ethanol increase can improve by 10% (conservative computation) in alcoholic industrialization, which is equivalent to increasing the income by RMB 600 yuan (6000 yuan/ton ethanol) from producing 1 ton of ethanol. The net revenue can increase to RMB 525 Yuan (600-75) from producing 1 ton of ethanol. The fuel ethanol production in Jilin Province is 0.6 million tons per year, while the fuel ethanol production in China is approximately 10.0 million tons per year in the near future. This technology can increase the net revenue of fuel ethanol in Jilin Province by RMB 3.2 billion Yuan, while the net revenue of fuel ethanol in China by RMB 52.5 billion Yuan per year. Moreover, the co-product will add a certain revenue due to the protein content increase of DDGS.

## 4. Discussion

Cellulolytic enzymes are believed to have common practical applications with the development of fuel ethanol production from lignocellulolytic biomass [28]. The application of cellulases is uncommon in the alcohol distilling industry, so a large amount of cellulose is unutilized. Few works have been conducted on the in situ conversion of corn fiber in alcoholic fermentation. 

The presence of non-starch polysaccharides, such as cellulose and hemicellulose in cereal grain, caused the excessive viscosity of the mash, which could make the process of starch hydrolysis and mash fermentation more complicated. In this work, we developed a novel lignocellulolytic enzymes preparation that had a strong substrate specificity for high-sugar and high-viscosity corn mash. These lignocellulolytic enzymes successfully promoted the ethanol yield from corn mash by more than 12% on the premise of the same material amount. Use of the novel lignocellulolytic enzyme preparation decreased mash viscosity, facilitating the technological process and leading to the release of certain amounts of glucose. This in turn provided an opportunity to increase ethanol efficiency. A quarter of the ethanol yield increase was due to the further hydrolysis of starch, while three quarters were to the hydrolysis of cellulose. 

The reported work showed lignocellulolytic enzymes could improve ethanol yield from starch due to excessive enzyme dosage. Table 5 presents a comparison of the ethanol fermentation from starch under conditions similar to those in our studies. The previous patent showed the addition of the cellulolytic enzyme can increase ethanol yield by as much as 10% or more at the enzyme dosage in the range of 0.01% to 0.1% wt/wt (enzyme protein/mash solid) [29]. The excessive enzyme input brings out an inappropriate ethanol yield increase, which is unacceptable and hinders cellulase application in the field of starch bioethanol. In our work, the novel lignocellulolytic enzymes cocktail could improve ethanol yield by more than 12% at the enzyme dosage of 0.0037% wt/wt (enzyme protein/mash solid), which is lower in cost and more economical than previous work. 

This patent by Abbas et al. showed that the ethanol yield from corn after adding cellulolytic enzymes from *T. reesei* was 95.7 g/L at the solid concentration of 28% (*w*/*w*), while Klosowski et al. reported that when corn mash is digested with a multi-enzyme complex from *Aspergillus* sp., the ethanol concentration is 85.7 g/L [30]. Sapińska et al. used multi-enzyme complex *CeluStar XL* for corn starch and obtained a high fermentation yield of 104.9 g/L [6]. Wang et al. used glucoamylase from *Rhizopus* sp. for the SSF of raw corn flour. The conversion efficiency of raw corn flour to ethanol was 94.5% of the theoretical ethanol yield [6]. Li et al. used Genecor cellulase for the SSF and obtained the ethanol efficiency of 111.3 g/L [7]. Therefore, a longer fermentation time (48–72 h) than that in our study was needed for corn starch fermentation to ethanol [7,28,29], which was the maximum ethanol yield of 117.0 g/L in the 36 h fermentation period at the solid concentration of 27% (*w*/*w*). Overall, the ethanol yield from corn starch reported in this study was higher than that achieved by previous studies. The lignocellulolytic enzyme cocktail application produces lower enzyme dosage, shorter fermentation time, and more significant ethanol improvement, which is more economical than the previous report.

This technology does not require any additional equipment or control system and does not modify the alcoholic fermentation technology. The lignocellulolytic enzymes with excellent properties can improve economical benefit through minimal enzyme input in alcohol industrialization. This technology will bring a better understanding in engineering bioethanol improvement. The results provide new ideas for promoting fuel ethanol industrialization and the technological progress of renewable energy in China. 

## 5. Patent

This technology has been patented in China (patent no. 201811277689.7).

## Figures and Tables

**Figure 1 jof-08-00221-f001:**
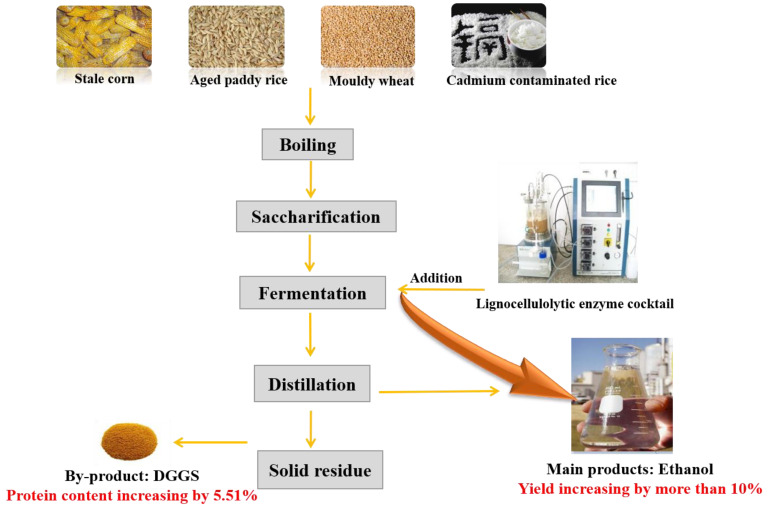
Further degradation of residual SSF broth after ethanol evaporated by lignocellulolytic enzymes cocktail.

**Table 1 jof-08-00221-t001:** Comparison of ethanol, reducing sugars, and viscosity at different enzyme loadings.

	20 FPU/L	30 FPU/L	40 FPU/L	45 FPU/L	50 FPU/L	60 FPU/L	80 FPU/L	100 FPU/L	150 FPU/L	200 FPU/L	CK
Ethanol (g/L)	105.1 ± 0.5	107.5 ± 0.7	112.7 ± 0.6	113.6 ± 0.6	114.2 ± 0.5	114.2 ± 0.7	114.3 ± 0.9	114.6 ± 0.7	113.8 ± 0.8	112.9 ± 0.7	100.7 ± 0.8
Residual sugar (g/L)	26.0 ± 1.4	21.1 ± 1.2	19.4 ± 1.0	18.1 ± 0.9	13.0 ± 0.7	13.3 ± 0.7	13.1± 0.8	13.2 ± 0.5	10.9 ± 0.4	11.1 ± 0.5	38.8 ± 1.7
Viscosity (Pa·s)	23.5	22.4	21.7	20.8	18.0	18.7	18.3	18.4	18.1	19.2	34.8

**Table 2 jof-08-00221-t002:** Comparison of ethanol, reducing sugars, and viscosity at different fermentation tanks.

Tank Size	0.3 L	1 L	5 L	70 L	Tank CK
Ethanol (g/L)	114.2 ± 0.3	113.8 ± 0.8	115.8 ± 0.5	117.0 ± 0.1	103.6 ± 1.0
Residual sugar (g/L)	16.0 ± 0.7	16.2 ± 0.5	16.4 ± 0.4	16.1 ± 0.1	36.8 ± 0.9
Viscosity (Pa·s)	18.0 ± 1.0	22.9 ± 1.2	21.6 ± 1.4	19.8 ± 1.1	33.5 ± 1.5

**Table 3 jof-08-00221-t003:** The composition analysis of unutilized corn mash after SSF and SHF with/without lignocellulolytic enzymes addition.

	SSF		SHF	
	With Lignocellulolytic Enzymes	Without Lignocellulolytic Enzymes	With Lignocellulolytic Enzymes	Without Lignocellulolytic Enzymes
Residual cellulose of corn mash (%)	7.48 ± 0.10	12.64 ± 0.52	10.07 ± 0.10	13.2 ± 0.52
Residual hemicellulose of corn mash (%)	5.07 ± 0.10	5.35 ± 0.10	5.74 ± 0.13	5.35 ± 0.10
Residual starch of corn mash (%)	5.34 ± 0.26	6.58 ± 0.86	5.06 ± 0.34	6.02 ± 0.86
Residual soluble sugar (g/L)	3.68 ± 0.86	6.42 ± 0.34	4.68 ± 0.56	6.11 ± 0.14

**Table 4 jof-08-00221-t004:** Composition of DDGS with/without lignocellulolytic enzymes addition.

Composition	With Lignocellulolytic Enzymes	Without Lignocellulolytic Enzymes	*p*-Value
Crude protein	29.63 ± 0.89	24.12 ± 0.73	<0.01
Cellulose	21.20 ± 0.42	33.12 ± 0.59	<0.01
Starch	11.26 ± 0.24	14.07 ± 0.33	<0.01
Ash	5.01 ± 0.11	5.48 ± 0.18	<0.01
Xylan	5.25 ± 0.13	8.21 ± 0.32	<0.01
Arabinan	1.89 ± 0.10	3.19 ± 0.17	<0.01
Water extractives	14.26 ± 0.52	6.01 ± 0.21	<0.01
Ether extractives	11.50 ± 1.02	5.80 ± 0.72	<0.01

**Table 5 jof-08-00221-t005:** Comparison alcohol yield under different fermentation conditions.

Enzymes/Microorganisms	Raw Sources	Solid Concentration (%)	Fermentation Conditions	Ethanol (g/L)	Reference
Lignocellulolytic enzymes from *T. reesei* and *A. niger*/*S. cerevisiae*	Corn	27.0	32 °C, SSF, 36 h	117.0	This study
Cellulolytic enzyme from *T. reesei*	Corn	28.0	30 °C, SSF, 45 h	95.7	[29]
Multi-enzyme complex from *Aspergillus* sp *enzymes*/*S. cerevisiae*	Corn	28.0	32 °C, SSF, 72 h	85.7	[30]
Multi-enzyme complex *CeluStar XL*/*S. cerevisiae*	Corn	28.5	30 °C, SSF, 72 h	104.9	[6]
Genencor cellulase/*S. cerevisiae*	Corn	27.0	30 °C, SSF, 48 h	111.3	[7]

## Data Availability

All data generated or analyzed during this study are included in this published article (and its additional file).

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
