# Peer review of "In Situ Corn Fiber Conversion for Ethanol Improvement by the Addition of a Novel Lignocellulolytic Enzyme Cocktail"

_jof, 2022, doi:10.3390/jof8030221_

Round 1

Reviewer 1 Report

I reviewed the manuscript entitled, In-situ corn fiber conversion for ethanol improvement by the addition of novel lignocellulolytic enzyme cocktail. The introduction is well written. Several revisions are required before final acceptance.

Methodology

All the methodology should be provided with detailed information, such as sections 2.5 and 2.7. Section 2.7 is just provided with reference. A detailed methodology should be added.

Very poor methodology. Authors should consider to provide methodology in detail

Table 4. statistical analysis must be conducted

The discussion is very poor and I suggest improving it in detail. Findings are interesting  

References are not according to journal format. Please revise it.

Find the attached PDF

Author Response

Reviewer 1: I reviewed the manuscript entitled, In-situ corn fiber conversion for ethanol improvement by the addition of novel lignocellulolytic enzyme cocktail. The introduction is well written. Several revisions are required before final acceptance.

Q1:All the methodology should be provided with detailed information, such as sections 2.5 and 2.7. Section 2.7 is just provided with reference. A detailed methodology should be added.Very poor methodology. Authors should consider to provide methodology in detail.

A1: Thank you very much. Some description has been added in the revised manuscript in red.

Q2:Table 4. statistical analysis must be conducted.

A2: Thank you very much. The error bar has been added in the revised manuscript in red.

Q3:The discussion is very poor and I suggest improving it in detail. Findings are interesting.

A3: Thank you very much. The discussion has been revised in the the revised manuscript in red.

Q4: References are not according to journal format. Please revise it.

A4: I am very sorry. The format of references has been revised in the manuscript.

Reviewer 2 Report

The article "In-situ corn fiber conversion for ethanol improvement by the addition of novel lignocellulolytic enzyme cocktail" is very interesting and written well. The article discussed about increasing the ethanol yield of corn mash by using a novel lignocellulolytic enzyme cocktail. Also in this article they showed that the enzyme cocktail can decrease the viscosity of corn mash. In addition, they showed that the use of novel cocktail enzyme can enhance the ethanol yield by 10%.

Introduction was written well for the article by referring recent research publications. Also described the methods well, but they need to write the methods more clearly. Results were presented in a good format. However there few suggestions to improve the article, hence I recommend the article for a publication after a minor revision. 

Minor points: 

Line 35: "a-amylase" change to "α-amylase".

Line 38-39: Any reference?, if yes, please provide it.

Line 76: Please provide more details about enzymes, where did you obtained and details.

Table 1: Is there a significant difference in the ethanol yield between the 40 FPU/L and 50 FPU/L enzyme loadings, please do a significance test.

Similarly in Table2, the difference between the ethanol yields?, please do a significance test to see whether they are significantly different.

Round 2

Reviewer 1 Report

Authors failed to answer the questions raised by me.

question : Table 4. statistical analysis must be conducted. Authors answer was: The error bar has been added in the revised manuscript in red. This is not the answer. Statistical analysis (significant difference) must be conducted; Error bar is nothing to do with significant difference.

Author Response

Feb  17, 2022

Dear Editor of Journal of fungi,

Thank you very much for your comments on manuscript No. jof-1606809R1. Those comments are all valuable and helpful for revising and improving our paper.  We have modified the manuscript according to your comments. The revisions are marked in red in marked manuscript. The point-by-point answers to the comments and suggestions are listed below.

Reviewer 1: Authors failed to answer the questions raised by me.

Q1: Table 4. statistical analysis must be conducted. Authors answer was: The error bar has been added in the revised manuscript in red. This is not the answer. Statistical analysis (significant difference) must be conducted; Error bar is nothing to do with significant difference.

A1:  I am very sorry for my mistakes.  The statistical analysis of data in Table.4 has  been added in the section 2.8 and section 3.5 in the revised manuscript in red.

Again, thank you very much. We are grateful for the time you spent reviewing and editing our manuscript. We greatly appreciate it.

The manuscript has been resubmitted to your journal.

We look forward to your positive response.

Best regards,

Sincerely yours,

Le Gao, Ph.D/ Associate Professor

Tianjin Key Laboratory for Industrial Biological Systems and Bioprocessing Engineering, Tianjin Institute of Industrial Biotechnology, Chinese Academy of Sciences, Tianjin, China

Email: gao_l@tib.cas.cn

Tel: +86-22-24828745